# Preserving Privacy of High-Dimensional Data by *l*-Diverse Constrained Slicing

**Zenab Amin [1], Adeel Anjum [2], Abid Khan [3,*], Awais Ahmad [4] and Gwanggil Jeon [5,*]**

1. Department of Computer Science, COMSATS University Islamabad, Park Road, Chak Shahzad, Islamabad 44000, Pakistan; zenabameen@gmail.com
2. Institute of IT, Quaid-i-Azam University, Islamabad 45320, Pakistan; aanjum@qau.edu.pk
3. School of Computing and Engineering, University of Derby, Derby DE22 1GB, UK
4. Department of Computer Science, Air University, Islamabad 44000, Pakistan; aahmad.marwat@gmail.com
5. Department of Embedded Systems Engineering, Incheon National University, Incheon 22012, Korea
* Correspondence: a.khan3@derby.ac.uk (A.K.); gjeon@inu.ac.kr (G.J.)

**Abstract:** In the modern world of digitalization, data growth, aggregation and sharing have escalated drastically. Users share huge amounts of data due to the widespread adoption of Internet-of-things (IoT) and cloud-based smart devices. Such data could have confidential attributes about various individuals. Therefore, privacy preservation has become an important concern. Many privacy-preserving data publication models have been proposed to ensure data sharing without privacy disclosures. However, publishing high-dimensional data with sufficient privacy is still a challenging task and very little focus has been given to propound optimal privacy solutions for high-dimensional data. In this paper, we propose a novel privacy-preserving model to anonymize high-dimensional data (prone to various privacy attacks including probabilistic, skewness, and gender-specific). Our proposed model is a combination of *l*-diversity along with constrained slicing and vertical division. The proposed model can protect the above-stated attacks with minimal information loss. The extensive experiments on real-world datasets advocate the outperformance of our proposed model among its counterparts.

**Keywords:** business intelligence; privacy-preserving data publication; high-dimensional data; *l*-diversity; constrained slicing

## 1. Introduction

With the advent of cloud computing, Internet-of-things (IoT) and smart devices, there has been a drastic uplift in data growth, aggregation and sharing. Large amounts of data are shared over various applications daily, e.g., social media, emails, cloud-based systems, and smart devices. These smart devices are very helpful in collecting and sharing electronic health records (EHRs) [1–3]. These shared data are very useful for researchers or data users to perform various types of statistical analysis, decision and policy making, conducting surveys, extensive research and general learning. The data researchers or users can be a simple reader to a medical doctor, a pharmaceutical expert, a governmental organization or any other entities seeking the data. In the context of a shared healthcare dataset, the data are attributed to various individuals who are known as data owners. The hospital which aggregates the data is known as the data holder. The shared healthcare dataset may contain different attributes of the data owners' data, e.g., name, age, zip code, gender, disease, diagnosing criteria, physician, treatment, etc. Certain above-stated attributes are sensitive in nature and the data owners would less likely intend to get them published or revealed, e.g., disease, diagnostic criteria, physician, treatment, etc. The insecurity of data owners about the sensitive attributes' disclosures is completely [4,5] genuine because these attributes' disclosures may somehow cause privacy leakage and discrimination. Therefore, the privacy preserving of such data has emerged as a vital need

of the modern digitalized world. However, if the data are made more private, the utility of the data reduces. Privacy-preserving data publication aims at setting an acceptable trade-off between data privacy and data utility. So, the high utility data may be published with acceptable privacy. These datasets consist of various types of attributes of data owners, e.g., explicit identifiers (name, patient ID, etc.), quasi-identifiers (age, gender, zip code, etc.) and sensitive attributes (disease, symptoms, treatment, etc.). The sensitive attributes contain highly sensitive and personal information related to the data users, which can be breached by some potential adversary. Due to their sharing process, security risks also arise during data transformation. However, security and privacy preservation is an essential part of the data sharing of individuals. The adversary with prior knowledge can relate certain attributes of published data (see Figure 1) with externally available datasets to target a victim. Therefore, data privacy has evolved as an inevitable need to the modern digital world. An authenticated survey indicated that 87 percent of the US population was vulnerable by mapping their different types of records with externally available datasets [6]. For protecting this sensitive data, many cryptographic techniques and models [7–10] have been proposed so far. However, the cryptographic models and techniques are generally computationally expensive. Apart from this, due to the sharing of keys to a limited number of known stakeholders, cryptographic models and techniques are not suitable for stakeholders at large (for example data publication for large and unknown stakeholders). On the other hand, the anonymization-based privacy-preserving data publication models and techniques are computationally inexpensive and suitable for data publication to large stakeholders. In recent and past decades, a plethora of privacy-preserving data publication models and techniques has been proposed [6,11–15]. The existing literature and models are not adequate to deal with high-dimensional data. As the number of nodes increases, the utility of anonymized high-dimensional data decreases and as the number of attributes increase, the chance of adversarial attacks also increases. In [16], the author presented a state-of-the-art privacy-preserving model for high-dimensional data. However, the proposed model resulted in a loss of information due to local recordings. We have focused on this flaw and improved the model in this work. The current work presents a novel privacy-preserving model, using *l*-diverse constrained slicing, to preserve the privacy of high-dimensional data publications with minimum or no loss of data. The proposed model vertically fragments the high-dimensional data on the basis of gender and applies *l*-diversity on either fragment to achieve anonymity. It improves the existing local recording approach and the utility of the data is preserved.

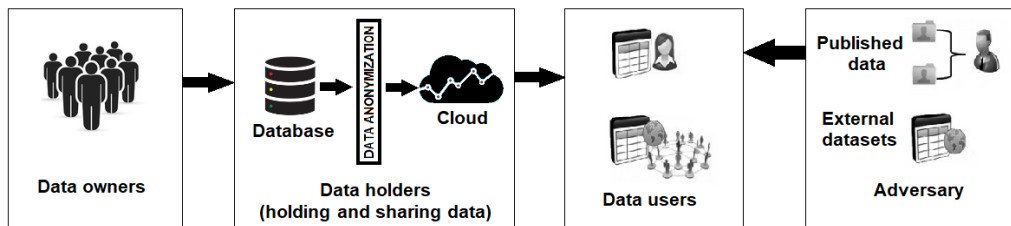

**Figure 1.** An overview of privacy-preserving data publication and adversarial attack.

The rest of the paper is organized as follows. Section 2 presents the literature review of the domain. Section 3 puts forward the preliminaries, problem setting and revisits the existing privacy model for high-dimensional datasets and its limitations. Section 4 presents our proposed model, its algorithms, methodology and protection analysis. Section 5 discusses the experimental evaluation based on evaluation matrices and finally, the paper is concluded in Section 6.

## 2. Literature Review

The privacy of the data belonging to data owners is a serious concern of today's digital world. On the other hand, the utility of the data is highly needed by data users and

researchers to make policies, analysis and decisions. An increase in one causes degradation in the other. Therefore, establishing a reasonable trade-off between enhanced utility and better privacy is a serious concern. Privacy-preserving data publishing is aimed at ensuring the provision of anonymized data to the data users. Several privacy-preserving models have been proposed so far in this regard. As discussed earlier, data have various types of attributes. Explicit identifiers are the types of attributes that identify the data owner uniquely and explicitly, e.g., ID, name, roll number, patient number, etc. Quasi-identifiers are the general attributes, e.g., age, zip code and gender, that may not help in identifying the specific data owner, although a combination of various quasi-identifiers (age = 20 years, zip code = 11,370, gender = male) may be exploited to identify the data owner. Sensitive attributes are the most confidential attributes that a data owner does not want to reveal, e.g., a disease. Initial researchers in privacy preservation removed the explicit identifiers from the publishable datasets so privacy could be achieved. However, an adversary could launch a linking attack on a group of quasi-identifiers with some externally available datasets (e.g., census or election voters' list). To overcome such privacy disclosure, the *k*-anonymity model was proposed [6], in which the concept of a quasi-identifiers group (QID-group) and equivalence class was presented. According to *k*-anonymity, each record in any of the QID-group is indistinguishable from other $k - 1$ records in the same QID-group. The main issues with *k*-anonymity were a utility loss due to generalization and the possibility of background knowledge and homogeneity attacks (a whole QID-group sharing the same sensitive attributes, e.g., dyspepsia). In [7], the authors addressed the homogeneity attacks and mitigated it by proposing *l*-diversity, which ensured the occurrence of *l* different sensitive attributes in a QID-group. However, the skewness and similarity attacks (a whole QID-group sharing the same type of sensitive attributes, e.g., gastroenterological like dyspepsia, acid reflux, stomach ulcer, etc. To overcome such issues, some researchers proposed [12] the t-closeness model, which ensured the distribution of sensitive attributes over the data bounded by a threshold t. However, this model was computationally expensive and somehow impractical to achieve. The above-mentioned privacy-preserving data publishing models were proposed to ensure data privacy with enhanced utility. However, the above-mentioned models and techniques are not applicable to high-dimension datasets, where sparseness of the data is huge [13,14]. There are two main reasons behind the inability of existing models to deal with high-dimensional data. First, the anonymization of high-dimensional datasets causes a huge information loss, which results in utility degradation [17]. Basically, the increase in the dimensions of the data means a higher occurrence and availability of associated attributes for the adversary to launch a correlation or linking attack. The second reason is that designing a suitable privacy-preserving solution for high-dimensional data with some prespecified requirements may not be possible. In the literature, it has been observed that some privacy-preserving generalization-based models [18–20] rely on the ambiguity between various data points within the specified spatial locality. As the sparseness of the data points increases, the distance between the points become less distinctive [21].

Some privacy-preserving models have been proposed to cater to the dimensionality issues. In [21], the author presented an LKC-privacy model for high-dimensional data. In the model, the author assumed that the adversarial prior knowledge was less than a threshold L. The above model assured that the maximum length of a set of quasi-identifier attributes would be published by at least k records. The model deals with L values in each iteration instead of potential quasi-identifier attributes. The problem with this model is its inability to handle large L values. In [22,23], techniques of feature selection and transformation were presented, respectively. These techniques have been widely used in dealing with high-dimensional data in data science. These techniques may not directly be applicable to the scenarios of privacy-preserving data publication because the data publishers are not aware of which attributes would be needed by the data users. Therefore, the selection of certain features may lead to a loss of data utility. It can be said that both

of the above-mentioned techniques usually result in transformed attributes with poor interpretability.

In [24], the author presented a state-of-the-art generalization-based privacy model for high-dimensional data. The model mitigated the risk of information loss caused by feature selection techniques through vertical partitioning of the data into two disjoint subtables. After vertical partitioning (classification), the author applied a heuristic method based on local recording on either subtable to ensure privacy. The smaller disjoint data tables were supposed to be anonymized separately in a more effective manner compared to high-dimensional data as a whole. If the anonymized data were published, the data users/researchers would be able to use part or all of it as per their requirements. Apart from this effectiveness, the main issue with this model was that the underlying local recording technique caused information loss due to generalization that could be mitigated by using some better privacy technique. A summarized overview has been illustrated in Table 1.

**Table 1.** An overview of previously proposed approaches.

| Ref. | Privacy Model | Techniques | Attacks |
|---|---|---|---|
| [6] | $k$-anonymity | Generalization + suppression | Background knowledge + homogeneity |
| [11] | $l$-diversity | Domain generalization | Similarity + Skewness |
| [12] | t-closeness | Generalization + suppression + SA-value distribution | Similarity |
| [13] | $p$-sensitive $k$-anonymity | Generalization + clustering | Homogeneity + similarity + linkage |
| [14] | p+sensitive $k$-anonymity | Generalization | Homogeneity + linkage |
| [14] | $(p/p_{+}, \alpha)$ sensitive $k$-anonymity | Generalization | Background knowledge + homogeneity |
| [15] | Balanced p+sensitive $k$-anonymity | Generalization | Skewness attack |
| [16] | High-dimensional data publishing via $k$-anonymity | Generalization | Attribute disclosure + high information loss |

## 3. Preliminaries

As a base for the upcoming discussion, we present some basic and conceptual definitions being used in the rest of the paper.

### 3.1. Basic Concepts

The raw microdata table (that is supposed to be published after anonymization) may be represented as T. In this notation, T has n number of person-specific records and m attributes. The m attributes can be categorized as follows on the basis of different degrees of openness.

- Explicit identifier: These are the attributes that may uniquely identify an individual/data owner in published information. The examples for explicit identifiers are ID, name, license number, social security number, etc.
- Quasi-identifying (QID) attributes: These are the general social attributes that can be used in a group (not independently) by an adversary to launch a privacy attack. The examples for quasi-identifiers are age, gender, height, weight, zip code, etc.
- Sensitive attributes: These are the attributes a data owner does not want to disclose or reveal publicly. A clear privacy leakage of someone's sensitive attribute may lead to discrimination, job issues, social embarrassment, or even life threats. The examples for

sensitive attributes are disease, salary, diagnostic criteria, symptoms, income, profit, loss, etc.

### 3.2. Formal Definitions

**Definition 1.** *(QID-groups) In a microdata table T, a QID-group refers to the group of records that share the same QID values. Alternatively, it is also known as EC. A table contains one or more QID-groups.*

**Definition 2.** *(Fragmentation) In the context of data science, fragmentation refers to a vertical partitioning of the publishable microdata table T into two or more disjoint subtables. Each subtable may be called a fragment.*

**Definition 3.** *(k-anonymity (k)) A microdata table T may be known as k-anonymous if and only if each record in the QID-group of T is indistinguishable by $k - 1$ other records in the same QID-group. Therefore, the probability of correct identification is at most $1/k$.*

**Definition 4.** *(l-diversity (l)) A microdata table T may be known as l-diverse if and only if all the records in the QID-groups of T are l-diverse.*

**Definition 5.** *(Slicing) A sliced table of a microdata table T is given by the tuple partition (see Definition 6) and attribute partition (see Definition 7).*

**Definition 6.** *(Tuple partitioning) For a table T having attributes t, the tuple partitioning comprises various subsets of t in such a manner that each tuple belongs to exactly one subset.*

**Definition 7.** *(Attribute partitioning) For a table T having tuples A, the attribute partitioning comprises various subsets of A in such a manner that each tuple belongs to exactly one subset.*

### 3.3. Revisiting a State-of-the-Art Privacy Approach for High-Dimensional Data

In [16], the author proposed a state-of-the-art privacy approach for high-dimensional datasets. The privacy approach had two phases: fragmentation and anonymization. The work handled the sparseness of the high-dimensional data by applying a straight vertical partitioning over them. The vertical partitioning referred to the process of fragmentation (see Definition 2). After fragmentation of the high-dimensional data, the author transformed them into two disjoint subtables called fragments. Each fragment was comprised of a smaller number of attributes having the same number of records. The second phase was the application of *k*-anonymity over the fragments to anonymize them. Through this, the burden of handling high-dimensional data decreased, while *k*-anonymity ensured classification accuracy. Apart from its effectiveness, this technique still faced attribute disclosure. Let us take the raw data in Table 2 and assume the application of these phases. The fragmentation phase divided the raw data table (Table 2) into fragment 1 and fragment 2. Afterwards, the *k*-anonymity was applied on each fragment. Tables 3 and 4 depict the anonymized fragments (fragment 1 and fragment 2, respectively). By relating the attributes of Tables 3 and 4, we may observe the attribute clerk in fragment 2, $ID = 9$, and $ID = 9$ in fragment 1 is working 36 h and has a marital status of married, is studying in junior sec. and their gender is female. The fragment 1 and fragment 2 tables clearly show that this scheme only restricts record linking attacks, not the attribute linkage attack.

**Table 2.** High-dimensional raw data.

| ID | Age | Gender | Education | Race | Marital Status | Family Members | Working Hours | Job | Class |
|----|-----|--------|-----------|------|----------------|----------------|---------------|-----|-------|
| 1 | 40 | M | Grad school | W | Married | 3 | 46 | Engineer | Y |
| 2 | 43 | F | Bachelors | A | Unmarried | 6 | 55 | Writer | N |
| 3 | 36 | M | Senior sec. | W | Unmarried | 2 | 48 | Engineer | N |
| 4 | 58 | F | Junior sec. | B | Married | 7 | 36 | Clerk | N |
| 5 | 32 | F | Senior sec. | W | Unmarried | 3 | 48 | Writer | N |
| 6 | 39 | M | Grad school | W | Married | 5 | 46 | Engineer | Y |
| 7 | 26 | F | Grad school | A | Unmarried | 5 | 60 | Writer | N |
| 8 | 27 | M | Grad school | W | Divorced | 2 | 48 | Engineer | Y |
| 9 | 45 | F | Junior sec. | W | Married | 6 | 36 | Clerk | N |
| 10 | 24 | F | Senior sec. | B | Unmarried | 4 | 36 | Clerk | N |

**Table 3.** Anonymized fragment 1 (satisfying 2-anonymity).

| ID | Age | Race | Family Members | Working Hours | Class |
|----|-----|------|----------------|---------------|-------|
| 1 | [27, 40] | W | [2, 5] | [46, 48] | Y |
| 6 | [27, 40] | W | [2, 5] | [46, 48] | Y |
| 8 | [27, 40] | W | [2, 5] | [46, 48] | Y |
| 2 | [26, 43] | A | [5, 6] | [55, 60] | N |
| 7 | [26, 43] | A | [5, 6] | [55, 60] | N |
| 3 | [32, 36] | W | [2, 3] | [46, 48] | N |
| 5 | [32, 36] | W | [2, 3] | [46, 48] | N |
| 4 | [24, 58] | * | [4, 7] | 36 | N |
| 9 | [24, 58] | * | [4, 7] | 36 | N |
| 10 | [24, 58] | * | [4, 7] | 36 | N |

* undefined data.

**Table 4.** Anonymized fragment 2 (satisfying 2-anonymity).

| ID | Gender | Education | Marital Status | Job | Class |
|----|--------|-----------|----------------|-----|-------|
| 1 | M | Grad school | Married | Engineer | Y |
| 6 | M | Grad school | Married | Engineer | Y |
| 8 | M | Grad school | Married | Engineer | Y |
| 2 | F | University | Unmarried | Writer | N |
| 7 | F | University | Unmarried | Writer | N |
| 3 | * | Senior sec. | Unmarried | * | N |
| 5 | * | Senior sec. | Unmarried | * | N |
| 10 | * | Senior sec. | Unmarried | * | N |
| 4 | F | Junior sec. | Married | Clerk | N |
| 9 | F | Junior sec. | Marriage | Clerk | N |

* undefined data.

## 4. Proposed Model

Our proposed model consists of two phases. In the first phase, we separated the raw data on the basis of gender. Afterward, we divided the separated raw data into small subsets of fragments using vertical partitioning. In the second phase, we applied the *l*-diversity with constrained slicing over the fragmented data. The detailed overview of our proposed model is as follows.

### 4.1. Vertical Fragmentation

In our proposed model, we are dealing with high-dimensional data. A plethora of recent work has shown that generalization results in information loss of a considerable amount, especially in the case of high-dimensional data. To reduce the dimensionality problem, we performed data fragmentation to get two smaller and disjoint fragments of

high-dimensional data. For this purpose, we took the raw data and divided them into different fragmented tables, while each fragment table maintained an equal proportion of attributes. Our partition metric considered the information required by the classification analysis. We made these fragments to enhance privacy and applied slicing in a more sophisticated manner. Fragmented tables contained similar attributes, such as the placement of QID attributes in the same fragmented table, while the placement of sensitive attributes was in the second fragmented table. Thus, the information that was required for classification should contain the high predictability of the sensitive attribute from attributes in each fragment. In our proposed model, first of all, we divided the raw data table into two parts. The division was done on the basis of gender (one table contained the males' data while the other contained the females' data) denoted by $T_m$ and $T_f$, respectively. Afterward, we took male and female data tables and applied fragmentation. Fragmentation was done on the basis of the similarity and correlation of QID and SAs attributes. We treated the entire non-identifier attributes as the QID attributes. The information l2 was defined to control the proportion of attributes in each fragment, denoted by $W$, which allocated attributes to each fragment. l2 could also be treated as the weight of $C_i$. It is worth noting that the value of l2 resided between 0 and 1; we should also limit the value of l1 to the same range such that l2 could be effective in balancing the QID attribute across fragments.

Therefore, if an attribute $A_i$ was assigned to a fragment $F_j$, the score of that assignment could be computed as:
Compute

$$(A_i, F_j) = (1 - P_j) \times (C_i - M_{ij}) \tag{1}$$

where,

$l_1 = (1 - P_j)$
$l_2 = (C_i - M_{ij})$
$P_j$: Proportion of attributes in $F_j$
$C_i$: Correlation between $A_i$ and *cls*
$M_{ij}$ = Average between $A_i$ and all attributes in $F_j$

$$
\begin{aligned}
P_j &= \frac{N_j}{N_{asi}} \\
C_i &= I(A_i, cls) \\
M_{ij} &= \frac{1}{|F_j|} \times \sum_{A \in F_j} I(A_i, A)
\end{aligned}
\tag{2}
$$

where
$N_j$: number of attributes in $F_j$;
$N_{asi}$: number of attributes assigned;
$l$(x, y): Correlation between attributes x and y
$|F_j|$: number of attributes in $F_j$.

At the start of the raw data partition, we assigned the class attribute to each fragment. Two attributes containing the highest value of correlation among them were selected as seeds and placed in the same fragments, such as F1 and F2. Then, we found the best iteration assigned at each fragment, described in Algorithm 1, whereby by repeating this process, we divided each fragment into multiple subfragments.

---

**Algorithm 1** Fragmentation of the raw data.

---

**Require:** D: raw data

    A: attributes $A_1$, $A_2$ ...

    *Cls*: class attribute in raw data

    S: set of attributes in raw data D

**Ensure:** F: fragmentation of raw data; divide into two fragments based on gender

  1:  **for** Genderdivision(Cls) **do**

  2:    F1 has males' data and F2 has females'

  3:    F1 ←(Cls)

  4:    F2 ← (Cls)

  5:  **end for**

  6:  F1 ← F1 U (Seed1)

  7:  F2 ← F2 U (seed2)

  8:  **while** S $\neq$ 0 **do**

  9:    **for** (f' $\in$ (F1, F2) **do**

10:      cur score score (Ai, Fi')

11:     **if** curscore > max score t **then**

12:       A' ← Ai

13:       F' ← Fi

14:       F ← F1, F2

15:     **end if**

16:    **end for**

17: **end while**F

---

*4.2. l-Diverse Constrained Slicing*

    Slicing is a novel technique for privacy-preserving data publishing. There are various advantages of slicing compared to generalization. It gives more utility than generalizing data. Slicing maintains more attribute correlations with the SAs. In slicing, we make columns having more than one attributes for a clear separation of QID and SAs. We used constrained slicing after fragmentation instead of generalization. In constrained slicing, we used random permutation. In this process, we created buckets by horizontal partitioning based on different ages. Moreover, we created columns having more than one attribute to reduce the dimensionality of the data. Membership disclosure and gender-based attacks were prevented using constrained slicing. In each subset, the authors placed the equivalence classes in one group, a process named bucketization, and then records were randomly shuffled to preserve the privacy in each bucket. We used *l*-diversity in replacement of *k*-anonymity to mitigate the information loss. The proposed approach was evaluated by using hospital-based datasets, and experiments showed that our *l*-diverse slicing model prevented the high-dimensional data from membership disclosure and gender-based attacks. We introduce a term known as constrained *l*-diverse slicing; this ensures that the privacy of the sensitive value of any individual with a probability greater than $\frac{1}{l}$ is protected from the adversary.

    Handling of bucketed/sliced data: in bucketization, as well as in slicing, attributes are split into two or more than two columns. For a bucket that contains t tuples and c columns, we generated m tuples as follows: We first randomized the values in each column. Then, we generate the *i*th tuple by linking the *i*th value in each column. We applied this procedure to all buckets and all of the tuples from the bucketed/sliced table. This procedure produced a linkage between the two columns randomly. From these tables, we are going to illustrate an example of slicing. First, we take the raw data (Table 5) from the cardio dataset after dividing the raw data based on gender. Table 6 contains the males' data and Table 7 consists of the females' data. Afterward, we fragment the males' data into two fragments by vertical partitioning: fragment one has quasi-identifiers (Table 8) and fragment two (Table 9) is related to sensitive attributes. Table 9 contains five QID attributes; gender, height, DOS, age and SC*P*-codes. Table 9 also has five sensitive attributes: weight, validated by human, HAXIS, hospital and the SC*P*- codes. Tables 10 and 11 have bucketed

data tables of males' fragment tables. We apply *l*-diversity and slicing on the bucketed data tables (Tables 10 and 11). *l*-diversity requires that every bucket has at least l sensitive values that are "well-represented" to ensure that each bucket contains at least l distinct values of the sensitive attributes. The increment of distinct, frequency values decreases the chances of a probabilistic attack Algorithm 2.

$$H(n) = -\sum_{s \in S} p(QID, s) log(p(QID, s)) \geq log(l) \qquad (3)$$

where *s* is known as a sensitive attribute, $P(QID, s)$ is the fraction of records in a QID-group possessing the sensitive value *s*. The left-hand side is known as the sensitive attribute's entropy, which possesses the property of sensitive group values that are equally distributed to produce large values. Hence, a larger threshold value indicates less certainty of inferring especially sensitive values in a group. It is noticeable that the inequality is not dependent on the selection of the log base. Assuming $p(s - t, B)$ as the probability that t takes sensitive value s provided that t is in bucket B, as explained by the law of total probability, the probability $p(t, s)$ is:

$$p(t, s) = \sum_B p(t, B) p(s|t, B) \qquad (4)$$

**Table 5.** High-dimensional raw data.

| Age | Gender | Height | DOS | Weigh T | Hospital | Heart-Axis | SC*P*-Codes | Validated by Human |
|-----|--------|--------|-----|---------|----------|------------|-------------|--------------------|
| 64 | F | 160 | 4/12/1994 | 74 | 1 | MID | NORM | TRUE |
| 68 | M | 180 | 4/12/1994 | 86 | 1 | LAD | IMI | TRUE |
| 66 | M | 170 | 3/12/1994 | 89 | 1 | MID | NORM | TRUE |
| 76 | M | 163 | 3/12/1994 | 51 | 1 | LAD | ASM | TRUE |
| 51 | M | 168 | 8/3/1995 | 72 | 4 | MID | NORM | FALSE |
| 80 | M | 177 | 8/3/1995 | 66 | 2 | MID | NST | TRUE |
| 75 | M | 158 | 9/8/1995 | 48 | 1 | ALAD | ASMI | FALSE |
| 64 | M | 165 | 9/8/1995 | 93 | 1 | ALAD | NDT | TRUE |
| 65 | F | 155 | 8/4/1996 | 70 | 1 | RAD | INJALLA | TRUE |
| 50 | M | 167 | 8/4/1996 | 75 | 8 | RAD | ASMI | TRUE |
| 55 | F | 170 | 3/5/1998 | 73 | 7 | NORM | ASMI | FALSE |

**Table 6.** Males' data table.

| Age | Gender | Heigh | DOS | Weight | Hospital | Heart-Axis | SC*P*-Codes | Validated by Human |
|-----|--------|-------|-----|--------|----------|------------|-------------|--------------------|
| 68 | M | 180 | 4/12/1994 | 86 | 1 | LAD | IMI | TRUE |
| 51 | M | 168 | 8/3/1995 | 72 | 4 | MID | NORM | TRUE |
| 64 | M | 165 | 9/8/1995 | 93 | 1 | ALAD | NDT | TRUE |
| 66 | M | 170 | 3/12/1994 | 89 | 1 | MID | NORM | TRUE |
| 80 | M | 177 | 8/3/1995 | 66 | 2 | MID | NST | TRUE |
| 76 | M | 163 | 3/12/1994 | 51 | 1 | LAD | ASM | TRUE |
| 75 | M | 158 | 9/8/1995 | 48 | 1 | ALAD | ASMI | TRUE |
| 50 | M | 167 | 8/4/1996 | 75 | 8 | RAD | ASMI | TRUE |

**Table 7.** Females' data table.

| Age | Gender | Height | DOS | Weight | Hospital | Heart-Axis | SC*P*-CODES | Validated by Human |
|-----|--------|--------|-----|--------|----------|------------|-------------|--------------------|
| 64 | F | 160 | 4/12/1994 | 74 | 1 | MID | NORM | TRUE |
| 55 | F | 170 | 3/5/1998 | 73 | 7 | NORM | ASMI | FALSE |
| 65 | F | 155 | 8/4/1996 | 70 | 1 | RAD | INJALLA | TRUE |

**Table 8.** QIDs of male data.

| Age | Gender | Height | DOS | SC*P*-Codes |
|-----|--------|--------|-----|-------------|
| 74 | 1 | MID | NORM | TRUE |
| 86 | 1 | LAD | IMI | TRUE |
| 89 | 1 | MID | NORM | TRUE |
| 51 | 1 | LAD | ASM | TRUE |
| 72 | 4 | MID | NORM | TRUE |
| 66 | 2 | MID | NST | TRUE |
| 48 | 1 | ALAD | ASMI | TRUE |
| 93 | 1 | ALAD | NDT | TRUE |
| 70 | 1 | RAD | INJALLA | TRUE |
| 75 | 8 | RAD | ASMI | TRUE |

**Table 9.** Sensitive attributes of males' data.

| BID | (Age, Gender, Height) | (DOS,SC*P*-Codes) |
|-----|-----------------------|-------------------|
| 1 | (31, M, 180)<br>(35, M, 168)<br>(33, M, 165)<br>(34, M, 170) | (4/12/1994, IMI)<br>(8/3/1995, NORM)<br>(9/8/1995, NDT)<br>(3/12/1994, NORM) |
| 2 | (56, M, 177)<br>(57, M, 163)<br>(60, M, 158)<br>(58, M, 167) | (8/3/1995, NST)<br>(3/12/1994, ASM)<br>(9/8/1995, ASMI)<br>(8/4/1996, ASMI) |

**Table 10.** *l*-diverse sliced males' QIDs table.

| Weight | Hospital | H-Axis | SC*P*-Code | Validated |
|--------|----------|--------|------------|-----------|
| 86 | 1 | LAD | IMI | TRUE |
| 72 | 4 | MID | NORM | TRUE |
| 93 | 1 | ALAD | NDT | TRUE |
| 89 | 1 | MID | NORM | TRUE |
| 66 | 2 | MID | NST | TRUE |
| 51 | 1 | LAD | ASM | TRUE |
| 48 | 1 | ALAD | ASMI | TRUE |
| 75 | 8 | RAD | ASMI | TRUE |

**Table 11.** *l*-diverse sliced males' SAs table.

| BID | (Weight, Hospital, H-Axis) | (SC*P*-Codes, Validated) |
|-----|----------------------------|--------------------------|
| 1 | (86, 1, LAD)<br>(72, 4, MID)<br>(93, 1, ALAD)<br>(89, 1, MID) | (IMI, TRUE)<br>(NORM, TRUE)<br>(NDT, TRUE)<br>(NORM, TRUE) |
| 2 | (66, 2MID)<br>(51, 1, LAD)<br>(48, 1, ALAD)<br>(75, 8, RAD) | (ASMI, TRUE)<br>(NST, TRUE)<br>(ASMI, TRUE)<br>(ASM, TRUE) |

---

**Algorithm 2** Fragmentation of the raw data

---

1: **Procedure**
2: **Input:** D: raw data
3: A: attributes $A_1, A_2, ...$
4: *Cls:* class attributes in raw data
5: S: set of attributes in raw data D
6: **Output:** F: Fragmentation of raw data
  */*divide a class on gender-base into two fragments*/*
7:    **for** *GenderDivision(Cls)* **do**
8:       $F_1$ is males' data and $F_2$ is females'data
9:       $F_1 \leftarrow (Cls)$
10:      $F_2 \leftarrow (Cls)$
11:   **end for**
  */*func two attributes having the maximum correlation Seed$_1$, Seed$_2$*/*
12:   $F_1 \leftarrow F_1 \bigcup (Seed_1)$
13:   $F_2 \leftarrow F_2 \bigcup (Seed_2)$
14:   **while**$(S \neq 0)$ **do**
  */* A′ is used to store the highest accurate attribute in each fragment.*/*
15:      **for**$(A \in S)$ **do**
  */*F′ is for best fragment and represent $F_1, F_2$*/*
16:         **for**$(F′ \in F_1, F_2)$ **do**
17:            cur score()
18:               **if** cur score $>$ max score $t$ **then**
19:                  $A′ \leftarrow A_1$
20:                  $F′ \leftarrow F_1$
21:                  $F \leftarrow F_1, F_2$
22:                  Return F
23:               **end if**
24:            **end for**
25:         **end for**
26:      **end while**
27: **end Procedure**

---

**Definition 8.** *Considering table $T(j)$, a QI is a subset of the tuples in $T(j)$. A part of $T(j)$ consists of disjoint QI groups whose union equals $T(j)$. Every QI group is allocated an ID which is unique in the partition. For every tuple t in $T(j)$, t.QI(j) denotes the QI group that contains t.*

**Definition 9.** *The anonymize version $T^*(j)$ of $T(j)$ is calculated based on a division of $T(j)$, and has several characteristics:*

1. *$T^*(j)$ contains a column $C_g$ called "GID" and all tuples in $T(j)$ except Cid.*
2. *Each tuple $t \in T(j)$ has tuple $t^* \in T*(j)$ such that $t^*[Cs] = t[Cs], t[Cg]$ is the ID of the hosting group of t in $T(j)$, and $t^*[C_iq_i](1id)$ is an interval covering $t[C_iq_i]$. The value of $t^*[C_iq_i]$ also proves Property 4.*
3. *For each QI, quasi identifier of $T(j)$, $T^*(j)$ might contain any tuples $t^*$ such that $t^*c[Cg]$ is a value in the domain of Cs, $t^*/c[Cg]$ equal to the GID of QI, and $t * [Ci//qi] (1 \leq i \leq d)$ is an interval to Property 4.*
4. *All attributes in $T^*(j)$ with the similar Cg have an similar value on QI attribute. These tuples make a quasi-identifier group in $T^*(j)$ whose ID is the Cg value in the group.*

*4.3. Protection Analysis*

In this subsection, a protection analysis of our proposed model is performed to evaluate the protection level against adversarial threats and the chances of occurrences of sensitive attributes' disclosures. The section explains how local bucketization acts in accordance with *l*-diversity without causing any huge information losses. Afterwards, the section proves that the local generalized and bucketized tables would also satisfy *l*-diversity by fulfilling

the necessary conditions. Initially, we prove the protection against the identity disclosures in a locally generalized table with the following lemma and corollary. *l*-diversity without causing huge information losses. Afterwards, we shall prove that the local generalized and bucketized tables would also satisfy *l*-diversity by fulfilling the necessary conditions. Initially, we prove the protection against the identity disclosures in a locally generalized table with the following lemma and corollary:

**Lemma 1.** *For a locally bucketized table T; having tuples* $t \in T$, *the occurrence probability of identity disclosures may be maximum as* $\frac{1}{G_{LE(t)}}$, *where* $G_{LE(t)}$ *is the local equivalence group having t tuples.*

**Proof.** As per the Definitions 1 and 4, all the tuples in a same $G_{LE(t)}$, share the same QIDs attributes values that are *l*-diverse. Therefore, an adversary would have to know at-least $G_{LE(t)}$ possible tuples after linking QIDs values, then the probability of occurring identity disclosure would be maximum $\frac{1}{G_{LE(t)}}$. □

**Corollary 1.** *A locally bucketized table may be l-diverse, if and only if every* $G_{LE(t)}$ *has at least l-diverse tuples.*

**Proof.** For a given locally bucketized table, in which each has at least *l*-diverse tuples. This means that, for any tuple $t \in T$, we have;

$$G_{LE(t)} \geq l \tag{5}$$

where, $|G_{LE(t)}|$ is the size of local equivalence group having *t*.
　　Then,

$$\frac{1}{G_{LE(t)}} \geq \frac{1}{l} \tag{6}$$

　　Now as per the Lemma 1, the occurrence probability of identity disclosure against any tuple *t* is maximum $\frac{1}{l}$. Hence it is proved that the complies with *l*-diversity.
　　Moreover, we discuss the level of protection for the sensitive attributes values in the locally bucketized table. Let's assume there is an adversary and he is aware of a tuple t's occurrence and its QIDs values. He tries to breach the privacy of that tuple by attempting to infer its sensitive attributes values. Then the adversary would need to match the QIDs values in the locally bucketized table. □

**Definition 10.** *(Linking tuples) For a given locally bucketized table* $T^{buc}$ *and any tuple t, a linking tuple lt exists such that if it is a linking tuple of t only if each and every QIDs values of t are that of lt.*

**Definition 11.** *(Linking buckets) For a given locally bucketized table* $T^{buc}$ *and any tuple t, a local bucket lb within the attributes is the linking bucket of t only if there is any linking attribute of t in lb.*
　　*We express the probability of exposure of sensitive value 'sv' of 't' as p(sv, t) and p(t, buc) as the probability of t being in the bucket buc. Then the lemma and corollary are as follows.*

**Lemma 2.** *For given a locally bucketized table* $T^{buc}$ *having t tuples, the probability of occurring a sensitive value 'sv'of t is expressed by following expression.*

$$p(sv, t) = \leq \sum_{lb} p(t, lb) \frac{lb(sv')}{mb} \tag{7}$$

　　*Where, lb(sv' is the number of most occurring sensitive value in the linking bucket lb, and* $|lb|$ *is the size of lb.*

**Proof.** For obtaining the sensitive values *sv*, the adversary needs to calculate the occurrence probabilities of *t* in each local bucket and *t* holding the sensitive values '*sv*' within each local bucket. Therefore, the adversary has;

$$p(sv, t) = \sum_B p(t, lb) p(sv|t, b) \tag{8}$$

where, $p(sv|t, b)$ shows the probabilities that *t* holding the sensitive values '*sv*' provided that the tuple '*t*' is in the local bucket *b*. The adversary skips the local bucket that does not hold any linking tuple of *t* as follows.

$$p(t, b) = o \, if \notin lt \in B \tag{9}$$

As per accordance with Definition 11, we have,

$$p(sv, t) = \sum_B p(t, lb) p(sv|t, b) \tag{10}$$

The most occurring sensitive value '*sv''* in *lb* is denoted as follows.

$$p(sv|t, lb) = \frac{lb(sv)}{lb} \le \frac{lb(sv')}{lb} \tag{11}$$

then,

$$p(t, sv) = \sum_{lb} p(t, lb) p(sv|t, lb) \le \sum_{lb} p(t, lb) \frac{lb(sv')}{lb} \tag{12}$$

□

## 5. Experimental Evaluation

In this section, we present the experimental evaluation of our proposed model. We implemented our proposed model on real-world datasets to validate the outperformance of our model in terms of better privacy and utility. Various evaluation matrices were used to check the proposed model's effectiveness concerning data utility and computational efficiency. In this regard, the results were comparatively analyzed with an existing state-or-the-art privacy model. The descriptions of the experimental evaluation is as follows.

### 5.1. Preparation and Setting

We implemented our proposed model in the Java programming language in the Eclipse internet development environment (IDE) on a machine having a Core i5 processor with 8 GB of RAM and a 1 TB hard disk. The operating system of the machines used was Windows 10. In our experimental evaluation, we had no special requirements for the dataset [25]. We were capable of processing any relational data with numerical and categorical attributes to check the effectiveness of our model. However, we evaluated our proposed model over hospital-based cardiology datasets. In our dataset, we collected the raw data of 21,000 entries. Afterwards, we applied fragmentation and then *l*-diverse slicing.

### 5.2. Evaluation Matrices

Data privacy is used to protect individuals' information from an adversarial attack. Several techniques have been introduced to ensure data privacy preservation. During this process, the actual data are distorted with respect to privacy aims, to secure individuals' information and to retain data utility. In this regard, we classified our microdata table into two fragments, already discussed in detail earlier. For the classification, we trained our classifiers, logistic regression (LR), support vector machine (SVM), and random forest (RF), using Python with *K*-fold cross-validation on the proposed dataset. The more similar and correlated classified data were stored in separate fragments, i.e., fragment 1 and

fragment 2. We compared the classifiers based on three evaluation metrics: accuracy, *K*-fold cross-validation, and ROC curves. The results are shown below.

### 5.3. Information Loss

Mostly information losses are measured on anonymized and actual datasets. In our proposed technique, we are not using any generalization to anonymize data. There is a low chance of information loss.

### 5.4. Counterfeits

Counterfeits play a vital role in data utility. Most of the existing data privacy techniques used counterfeits or dummy data to ensure data privacy. Our scheme does not use counterfeits to retain data utility.

### 5.5. Accuracy

Accuracy is one metric for evaluating classification models. Informally, accuracy is the fraction of predictions a model gets right.

$$H(n) = \sum_{s \in S} p(QID, s) log(p(QID, s)) \geq log(l) \tag{13}$$

For a binary classification, accuracy can also be calculated in terms of positives and negatives as follows:

$$Accuracy = \frac{TP + TN}{TP + TN + FP + FN} \tag{14}$$

where $TP$ = true positives, $TN$ = true negatives, $FP$ = false positives, and $FN$ = false negatives. Figures 2 and 3 shows the accuracy results of the three classifiers, where the accuracy of the LR and SVM are almost the same, while the RF performs the best. Table 12 shows the accuracy results of the three classifiers, where the accuracy of the LR and SVM are almost the same, while the RF has the best accuracy. K-folds: In general, to avoid over fitting we use cross validation. In standard *k*-fold cross-validation, we partition the data into *k* subsets, called folds.

$$TPR = \frac{TP}{TP + FN} \tag{15}$$

False Positive Rate (FPR) is defined as follows:

$$FPR = \frac{FP}{FP + TN} \tag{16}$$

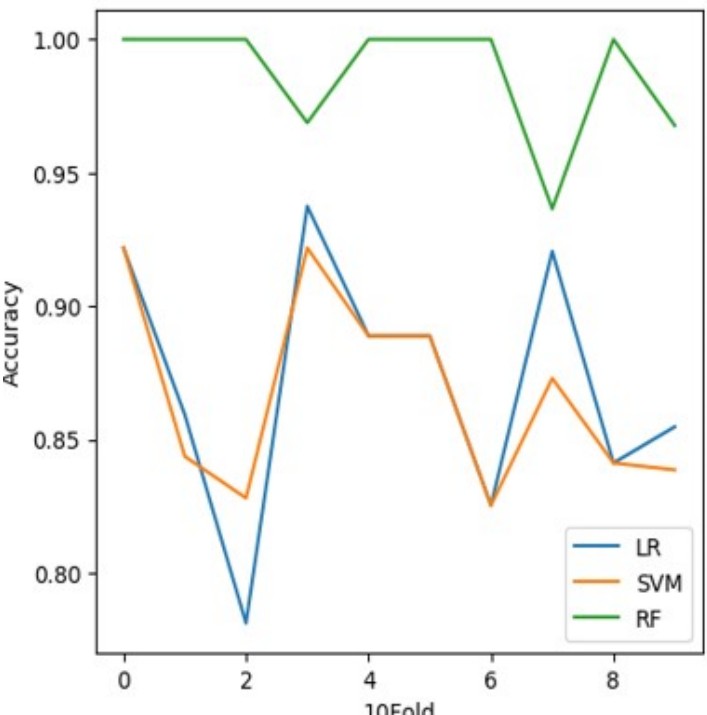

**Figure 2.** Comparison analysis of different classifiers over 10 folds.

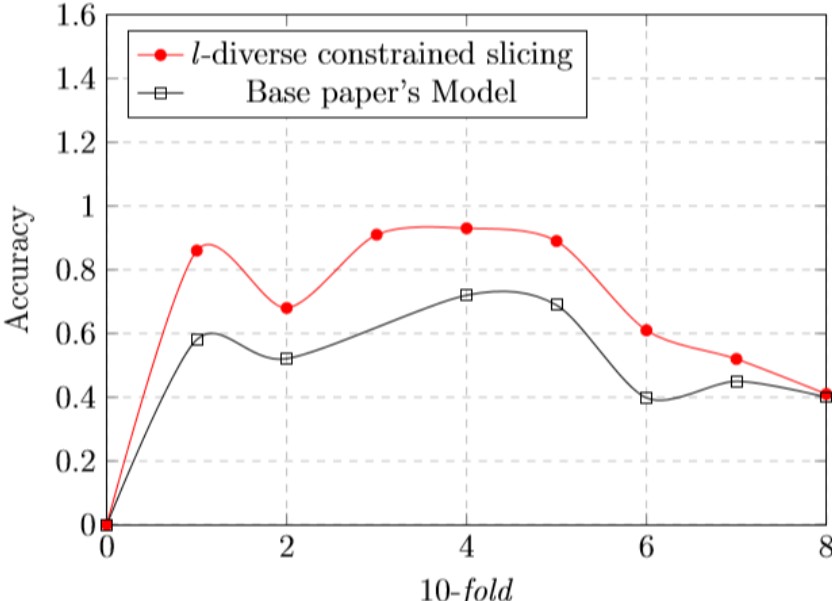

**Figure 3.** *Cont.*

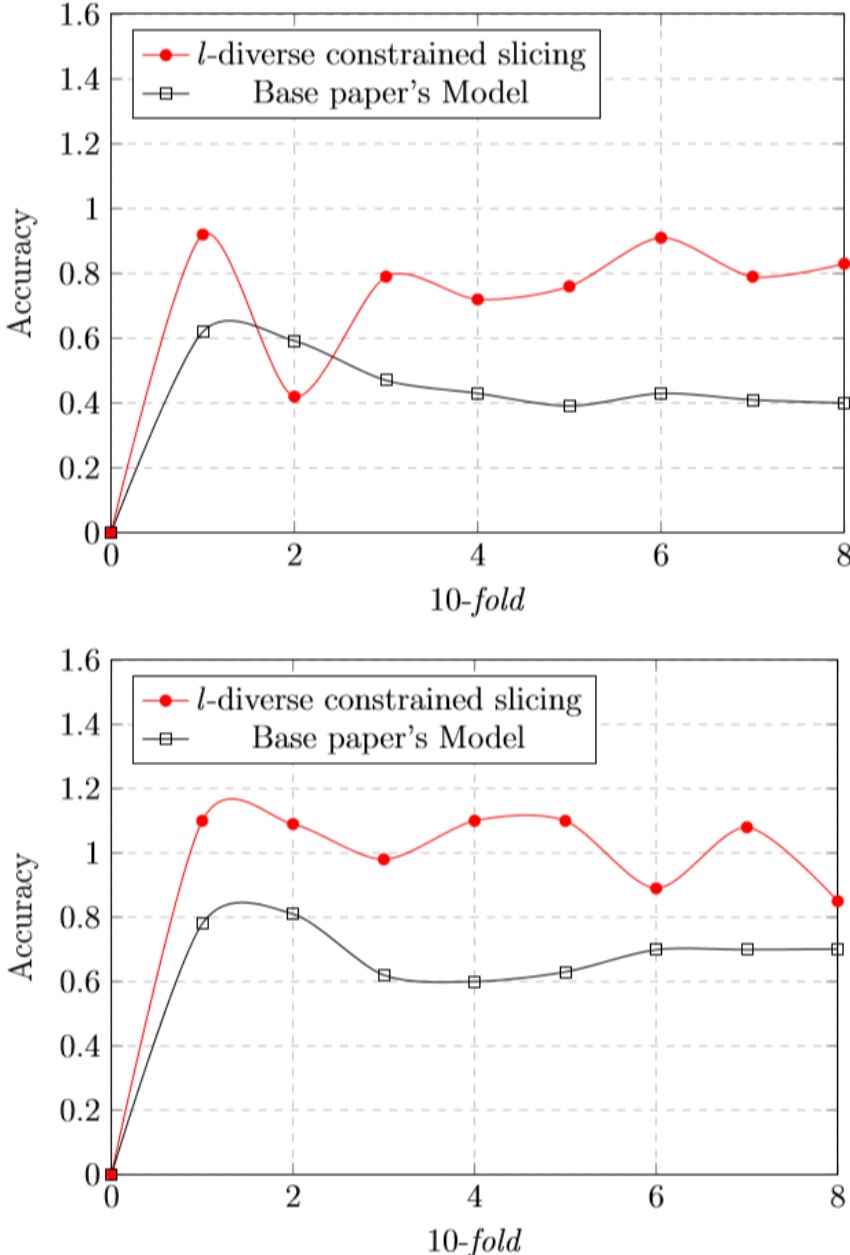

**Figure 3.** *l*-diverse constraint slicing and base paper's model comparison analysis of different classifiers (LR, RF and SVM) over 10 folds.

**Table 12.** Accuracy comparisons of different classifiers.

| Classifier | Accuracy in (%) |
|---|---|
| LR | 0.941 |
| SVM | 0.948 |
| RF | 0.992 |

After that, the algorithm is trained iteratively over $k - 1$ folds and the remaining fold is known as test set or holdout fold. Hyper-parameters are allowed to be tuned with the original training set by cross validation. It keeps the test set as a truly unseen dataset to choose the final mode. A comparison of these classifiers LR, SVM and RF has been depicted in Figure 3.

### 5.6. K-Fold Cross-Validation

In general, to avoid overfitting. we use cross validation. In standard *k*-fold cross-validation, we partition the data into k subsets, called *folds*. Then, we iteratively train the algorithm on $k - 1$ folds while using the remaining fold as the test set (called the "holdout fold"). Cross-validation allows us to tune the hyperparameters with only our original training set. This allows us to keep our test set as a truly unseen dataset for selecting our final model. Figure 2 shows the comparison of classifiers with 10 folds. Figure 3 shows the *l*-diverse constrained slicing and base paper model comparison of classifiers with 10 folds.

### 5.7. Training Test Accuracy of RF

We trained our support vector machine (SVM) and random forest (RF) classifiers using Python with a *K*-fold cross-validation on the above dataset to classify data based on similarity. The more similar and correlated data are put in the separate fragments, i.e., fragment 1 and fragment 2. To evaluate the accuracy of the classifiers, the F-score was adopted. The F-score, also called the F1-score, is a measure of a model's accuracy on a dataset. It is used to evaluate binary classification systems, which classify examples into "positive" or "negative". The F-score is a way of combining the precision and recall of the model, and it is defined as the harmonic mean of the model's precision and recall. The F-Score formula is:

$$F_1 = \frac{2}{\frac{1}{recall} \times \frac{1}{precision}} = 2 \times \frac{precision \times recall}{precision + recall} \qquad (17)$$

We have trained our dataset and tried to find the accuracy. Figure 4 shows the result of our proposed scheme and Figure 5 represents the accuracy graph of our proposed technique. The graph clearly shows that the accuracy of tested model is almost near to the trained model. Figure 6 depicts the comparison with our base approach. As can be seen, due to effective use of constrained slicing, our approach outperforms the base approach in terms of training and testing accuracy.

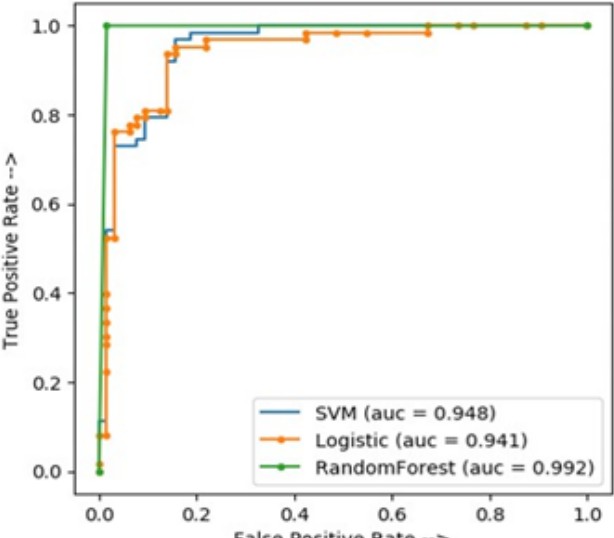

**Figure 4.** ROC curve comparisons of different classifiers.

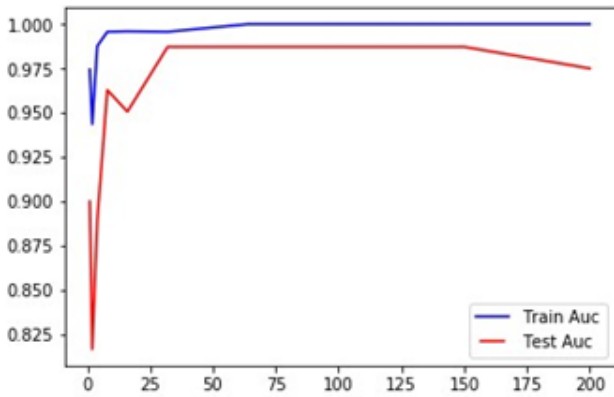

**Figure 5.** RF classification report.

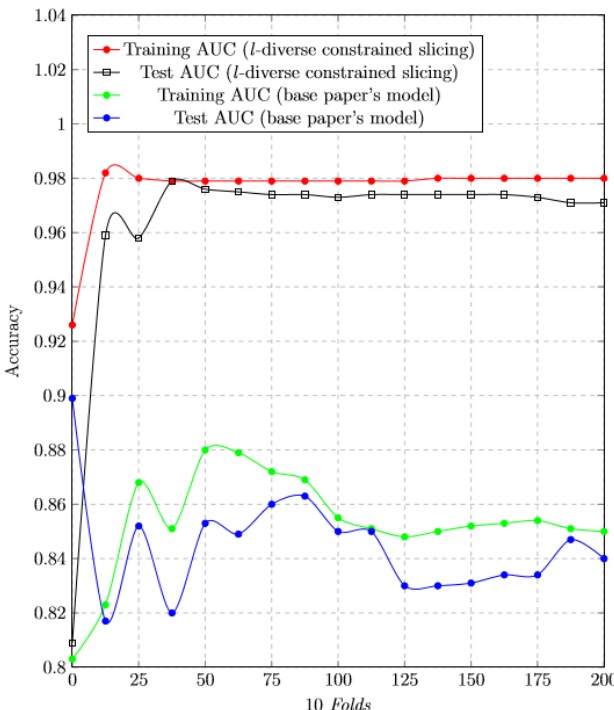

**Figure 6.** RF classification report—*l*-diversity constrained slicing graph comparison with base paper model.

We have trained our dataset and tried to find the accuracy. Table 13 represents the accuracy graph of our proposed technique. The graph clearly shows that the accuracy of tested model is almost near to the trained model.

**Table 13.** Training and test accuracy of RF.

| Precision | Recall | F1-Score |
|:---:|:---:|:---:|
| 0.97 | 0.95 | 0.98 |
| 0.95 | 0.98 | 0.97 |

### 5.8. l-Diverse Effect

As we know, *l*-diversity is used to protect sensitive attributes from an adversary attack. As the value of "*l*" increases, the probability of adversary attack decreases. Figure 7 represents the *l*-diversity constrained slicing graph comparison with the base paper model. With the increment of the bucket size, the level of diversity also increases. As the level

of "*l*" increases, the probability to infer records decreases. This is where our approach is well-suited to defend against attribute disclosure attacks.

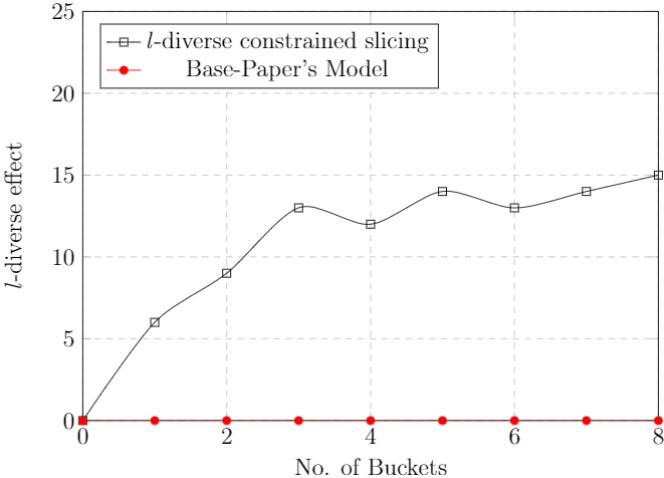

**Figure 7.** *l*-Diverse effect—comparison with base paper.

### 5.9. Time Cost

Time cost plays an important role in deploying any kind of scheme. To find the efficiency of a model or scheme, time is the main constituent. Figure 8 represents the time cost graph of our technique compared with that of the base paper. We notice that the time cost increases with respect to the data. If the amount of data increases, the time cost also increases.

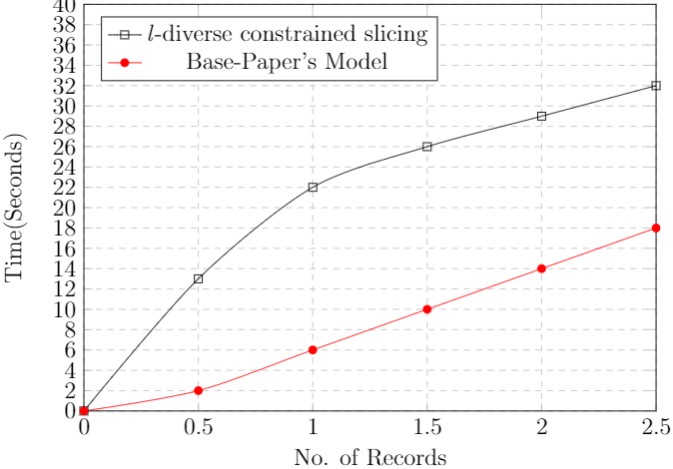

**Figure 8.** Time cost graph and comparison with base paper model.

### 6. Conclusions

Privacy-preserving data publication has shown widespread adoption in past and ongoing decades. However, establishing a reasonable trade-off between data utility and data privacy is still an open research question. Publishing data with high utility and enough privacy becomes a critical job when we are dealing with data having great sparseness and high-dimensional data. The anonymization of such high-dimensional data leads to a huge information loss. A few privacy-preserving techniques and models have been proposed so far for high-dimensional datasets. A state-of-the-art privacy model performed data fragmentation and applied local recording to anonymize fragments smaller than the whole high-dimensional dataset. The privacy model had some basic flaws; it was prone to attribute membership attack and furthermore, local recording caused a huge information loss. In

this paper, we mitigated these issues by dividing the high-dimensional data into gender-based datasets and applied *l*-diverse slicing instead of *k*-anonymity. Experimentation on real-world datasets showed that our proposed model outperformed a previous model in terms of privacy and utility.

**Author Contributions:** Conceptualization, Z.A., A.K. and G.J.; Data curation, Z.A.; Funding acquisition, Gwanggil Jeon; Methodology, Z.A., A.A. (Adeel Anjum), A.K., A.A. (Awais Ahmad) and G.J.; Project administration, A.A. (Awais Ahmad); Software, Z.A.; Supervision, A.A. (Adeel Anjum) and A.K.; Validation, A.K., A.A. (Awais Ahmad) and G.J.; Writing—original draft, Z.A.; Writing—review and editing, A.A. (Adeel Anjum), A.K., A.A. (Awais Ahmad) and G.J. All authors have read and agreed to the published version of the manuscript.

**Funding:** This research received no external funding.

**Institutional Review Board Statement:** This study was approved by the Institutional Review Board (IRB) of COMSATS University Islamabad and the protocols used in the study were approved by the Committee of Computer Science Department, COMSATS University Islamabad, Pakistan.

**Informed Consent Statement:** I have read and I understand the provided information and have had the opportunity to ask questions. I understand that my participation is voluntary and that I am free to withdraw at any time, without giving a reason and without cost. I understand that I will be given a copy of this consent form. I voluntarily agree to take part in this study.

**Data Availability Statement:** We have implemented our proposed model in Java Programming language in Eclipse internet development environment (IDE) on a machine; having Core i5 processor with 8GB RAM and 1TB Hard disk. The operating system of the machines used is Windows 10. Dataset: In our experimental evaluation, we have no special requirements for the dataset. We were capable of processing any relational data with numerical and categorical attributes to check the effectiveness of our model. However, we have evaluated our proposed model over cardiology hospital-based datasets. In our dataset, we collected the raw data of 21,000 entries. Afterwards, we applied fragmentation and then *l*-diverse slicing.

**Conflicts of Interest:** Authors state that there are no potential conflict of interest and will not consider those with competing interest.

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
