# Peer review of "Preserving Privacy of High-Dimensional Data by l-Diverse Constrained Slicing"

_electronics, doi:10.3390/electronics11081257_

Round 1
Reviewer 1 Report
The paper is well drafted . However , I have following concerns.1- The paper used keyword big data , however the authors failed to provide the applicability of purpose algorithm with big data .
2- The single paragraph introduction does not justified the existing problem . The author must clearly identify the problem and their contribution .
3- Author should have conciderd the most recent papers for literature that justify the problem .
4- Section 3 and 4 should be combined and avoid unnecessary details to provide easy of understanding for the readers .
5- Symbols used are algorithm 1 and 2 should be clear and concise . Also the symbols are confusing and algorithm steps requires refinement for cleariaty .
6- As most of the recently era generated data is catagarise as big data . The author should have conciderd multiple big data sets for experiments .
7- The paper must compare the results with recent published papers for at least two privacy loss and utility gain benchmarks .
8- The paper will look more scientifically sound if the paper provide time and space complexity for the proposed algorithms.
Reviewer 2 Report
This manuscript proposed a privacy-preserving model for high-dimensional data. This article has a broad interest in the computer science and engineering community.
The minor comments:
- the math notation in the paragraph is better to use math mode, so they can easily differ from the other words;
- Line 82, the content in "(age = 20 years, zip-code = 11370, gender = male)" is not necessary. Just mention the combination of "age, zip-code and gender...";
- Line 110: "In the litterateur", type error;
- Line 145: "may be represented as .", missing notation;
- Line 170-171: better to explain what is the l-diverse;
- page 7: ( , ) = (1 where l1 = (1 l2 = (Ci ) : Proportion of attributes in Ci: Correlation between and cls. the number of "(" and ")" does not match. It is better to explain the meaning of the formula.
Reviewer 3 Report
I do like the consolidation nature of the paper.
However, the authors should discuss the limitations and not only positive light --> they only relate to a certain class of attacks but may discuss the informative nature in general the reduction in privacy and information lost in general and where still information can be gained with the proposed scheme.
A second however, I have several issues with formality and notations etc.:
1 line 145 --> missing formula "may be represented as "XXX". " --> correct.
2 line 170 --> l-diversity should be formally defined (the background notation and reference are not enough).
3 some problem as line numbers are missing in pages 6-7.
4 subscripts are needed all over the paper --e.g. Ci etc. inline text
5 Definition of Eq (1) and immediately after text--> is all fregmanted and not well defined and clear. --> all internal brackets in the line starting with `( , )` are wrong.
6 Eq. 2 C_i defined from C_i --> should have a different notation
7 algorithms are with issues: (e.g.) no spaces whileF, (e.g.) no closing brackets (f' \in (F1,F2). <-- please also add subscripts in all places where needed.
8 Entropy definition --> don't you miss a minus ?
9 Algorithm 2 - all \list or \STATE elements to make it organized and not just a blurb of text. Discuss as a pseudo algorithm and not as a full fledged algorithm.
10 also wrong statements: "if p (t, s) 1/l true then" in the algorithm --> fix thoroughly all !!
11 line 224 --> Ciqi --> use sub/super scripts --> inline equations are not OK and not nicely readable
12, 13 --> Figures 2 and 3 4 --> not OK with not a bad quallity regenerate and rescale.
14 IMO recall F1 TPN etc. should not be there as it is basic (perhaps only as one liner short formulas ...)
Round 2
Reviewer 1 Report
The authors try to justify the work, however, there are many issues with the current version of the paper.
- The author should list and compare in a table “the most relevant and recent papers” that correlate with their study and clearly identify the base paper and its reason other than table 1.
- The symbols and notation should have a tabular listing to facilitate the readers. Some notations are used in the algorithm is confusing to understand such as Mij.
- If the authors used the existing data set it should be cited. If not, the data processing section should be added to explain the steps about data.
- The results reported in figures (2,3,4,5, and 6) does not compare the proposed work with existing studies, The author should use standardized matrices to compare multiple data sets and existing study results.
- The author should provide at least two utility and privacy matrices in comparison with counterparts.
- The figures (2,3,4,5, and 6) should be revised so that it is clear and in the same format.
Reviewer 3 Report
(1) Figures - were just rescaled, Quallity is still very bad ...
(2) Equations and notations:
(a) cannot understand the notations here - (Ai, Fj) = (1 Pj) (Ci, Mij) ?
with comma, without a comma - define the operation element mult. / vector mult. / concatenation etc ...
(b) also here - either you define it well or fix it: l1 = (1 l2 = (Ci )
a missing closing bracket ? the operation performed etc.
(3) line 224: AFj Where - AFj is just hanging there ?
(4) define operation and brackets () [] what does each define ..
